# Resolvins’ Obesity-Driven Deficiency: The Implications for Maternal–Fetal Health

**DOI:** 10.3390/nu14081662

**Published:** 2022-04-16

**Authors:** Alice Bosco, Angelica Dessì, Caterina Zanza, Roberta Pintus, Vassilios Fanos

**Affiliations:** Department of Surgical Sciences, University of Cagliari and Neonatal Intensive Care Unit, AOU Cagliari, 09124 Cagliari, Italy; alicebosco88@gmail.com (A.B.); caterina.zanza92@gmail.com (C.Z.); gomberta@icloud.com (R.P.); vafanos@tiscali.it (V.F.)

**Keywords:** resolvins, obesity, metabolomics, breast milk

## Abstract

Since pregnancy is already characterized by mild but significant inflammatory activity in physiological conditions, when complicated by obesity the probability of a persistent inflammatory state increases, with consequent multiple repercussions that add up to the complications associated with acute inflammation. In this context, the role of resolvins, specialized pro-resolving mediators (SPMs), deriving from omega-3 essential fatty acids, may be crucial. Indeed, differential production in numerous high-risk conditions associated with both childbirth and neonatal health, the correlation between maternal omega-3 intake and resolvin concentrations in maternal blood and at the placental level, and the high values found in breast milk in the first month of breastfeeding, are some of the most important hallmarks of these autacoids. In addition, a growing body of scientific evidence supports the lack of SPMs, at the level of immune-metabolic tissues, in the case of obesity. Furthermore, the obesity-related lack of SPMs seems to be decisive in the context of the current outbreak of COVID-19, as it appears to be one of the causes associated with the higher incidence of complications and negative outcomes of SARS-CoV-2 infection. The usefulness of metabolomics in this field appears clear, given that through the metabolome it is possible to observe the numerous and complex interactions between the mother, the placenta and the fetus in order to identify specific biomarkers useful in the prediction, diagnosis and monitoring of the various obstetric conditions. However, further investigations are needed in order to evaluate the possible use of some resolvins as biomarkers of maternal–fetal outcomes but also to establish adequate integration values in pregnant women with omega-3 fatty acids or with more active derivatives that guarantee optimal SPM production under risky conditions.

## 1. Introduction

Pregnancy represents a great opportunity to secure the future health for both the mother and the offspring [1,2,3]. In fact, the negative impact on the offspring of some maternal metabolic alterations, such as obesity and diabetes, is now known, primarily due to the teratogenic effect at the embryonic level but also due to the greater probability of preterm birth and stillbirth. Furthermore, to date, there is substantial scientific evidence regarding a possible metabolic programming of the offspring with longer-term effects, including an increased risk of developing metabolic pathologies and cardiovascular problems [2,3]. As for the pregnant woman, she is faced with important physical changes and metabolic adaptations aimed at protecting and guaranteeing correct fetal development, the result of which is in close correlation with the pre-pregnancy nutritional state and with weight gain during gestation [1]. Therefore, there is a growing concern for the recent increase in the number of women who become pregnant with a body mass index (BMI) ≥30 kg/m^2^ [4]. In fact, even from the perspective of maternal health, as gestational obesity is associated with an increase in mortality and morbidity not only in the fetus but also in the mother. Indeed, severe complications for the obese pregnant woman include preeclampsia, thromboembolism and an incidence of gestational diabetes (GDM) up to 3.8 times higher [2,3,5]. Obesity in pregnancy is also responsible for consequences on maternal health even in the long term, with an increase in the incidence of cardiovascular and metabolic problems.

In addition, there is the negative impact of the SARS-CoV-2 pandemic on eating habits. The most recent scientific literature [6] has in fact highlighted how the lockdown due to the COVID-19 pandemic was characterized by a worse eating behavior and a reduction in physical activity in the population, with great difficulties in weight management. These problems were even more important in subjects with high BMI values, who were characterized by the lowest levels of physical activity, the worst dietary quality and the highest rate of consumption of excessive quantities of food [6]. These data are in agreement with the negative influence of the SARS-CoV-2 emergency on the glycemic control of pregnant women [7,8]. Furthermore, the COVID-19-related decline in mental health also had severe repercussions on eating behavior, predicting over-eating and less movement [6]. Specifically, it was found that emotional eating affected several pregnant women during the COVID-19 pandemic and this caused excessive weight gain during pregnancy [9]. 

Moreover, obesity in pregnancy and gestational diabetes are frequently associated with a chronic low-grade inflammation, defined as meta-inflammation [3], whose impact on the possible onset of maternal–fetal complications is now established [10]. The altered response to pathogens caused by meta-inflammation was already known prior to the SARS-CoV-2 epidemic [11], and recent scientific evidence on this subject has shown that this chronic low-grade inflammation appears to exacerbate the underlying pathogenetic mechanisms of COVID-19 through several mechanisms, including an impairment of innate and adaptive immune responses and an increase in chronic inflammation and oxidative stress [12]. In addition, there are the data on the higher incidence of complications in the premature newborn subjected to a pro-inflammatory intrauterine environment, including necrotizing enterocolitis, paraventricular leukomalacia, intraventricular hemorrhage, pulmonary complications and retinopathy [13,14,15].

In recent years, the scientific literature [10] has highlighted how particular lipid mediators of inflammation, called specialized pro-resolving mediators (SPM), deriving from omega-3 polyunsaturated fatty acids (PUFA), are produced in a different way under numerous high-risk conditions associated with both childbirth and neonatal health.

The correlation of these data with the important contribution of omega-3 LC-PUFAs for maternal–fetal health [16], including the reduction in the risk of preterm birth before 34 weeks, and for their role as modulators of chronic low-grade inflammation [17], underlines the centrality of derivatives of omega-3 fatty acids in this context. Furthermore, there is numerous experimental evidence, both in vitro and in vivo, to support that the increase in SPM following dietary supplementation with omega-3 may be at the origin of the observed benefits [10,18,19,20,21].

## 2. Acute and Chronic Inflammation

The inflammatory response is essential for the body’s defense against a pathogenic infection or a harmful insult [10,22]. Specifically, there is a local reaction of the vascular system following the alteration of homeostasis which consists of an increase in the permeability and perfusion of blood vessels. This mechanism, which is activated in a few minutes, is necessary to allow the extravasation of circulating leukocytes and some plasma proteins, involved both in the disinfection of tissues and in the regulation of the inflammatory process itself and of the antigen-specific immune response. In this context, particular eicosanoids, or lipid mediators of inflammation, play a key role, the main ones being some derivatives of arachidonic acid (AA) such as leukotrienes and prostaglandins [10,18]. The formers mainly have chemotactic properties for neutrophils and regulate vascular permeability, while prostaglandins are the main actors responsible for changes in blood flow (favoring leukocyte migration and plasma exudation to damaged tissue). However, the inflammatory response is self-limiting in its acute form [19]. In fact, this purely pro-inflammatory phase is followed by another one. This normally leads to tissue and homeostasis restoration, thanks to some fundamental physiological processes that limit both the extension and the duration of the inflammatory response [20]. In the first phase, there is both the regulation of the granulocyte recruitment rate and the control over their state of activation, while in the following purely resolutive phase, there is the removal of the granulocyte infiltrate in order to restore the normal functions of the damaged tissue [10,18]. The control of this phase occurs at two levels: systemic and local [19]. The systemic level involves both circulating glucocorticoids and the acute phase response [10] and some anti-inflammatory efferent cholinergic neuronal pathways [21]. In the latter case, the regulation of tissue-specific reactions is necessary to limit inflammatory lesions and restore homeostasis.

Although numerous components of the classical inflammatory response are involved, such as a systemic increase in circulating inflammatory cytokines and acute phase proteins (C-reactive protein), leukocyte recruitment to inflamed tissues and activation of tissues together with reparative responses of the tissues themselves, the inflammation characteristic of obesity shows unique and peculiar traits [23]. First of all, it is chronic in nature [24] and is mainly triggered by nutrients and energy surplus [25,26]. Furthermore, in this specific condition, the inflammatory molecules and the metabolic pathways involved play a dual role, both as inflammatory mediators and as regulators of metabolism and energy storage [27]. In addition, the characteristic chronic nature determines a low-grade tonic activation of the innate immune system responsible for the alteration of metabolic homeostasis over time [23,24]. This mainly affects adipose tissue where there is an increased production of pro-inflammatory adipokines including tumor necrosis factor (TNF) α, interleukins (IL)-1ß, -6 and leptin and resistin [25,28]. Evans et al. [29] also found that the increase in the production and release of inflammatory adipokines is proportional to the amount of adipose tissue present. At the same time, a decreased production of anti-inflammatory factors was recorded, such as adiponectin [26,28] and an intrinsic inability to resolve inflammation and restore homeostasis and tissue function [27]. Secondly, the multi-organ repercussions of obesity-related inflammation are caused by its distinctive features as they are not normally associated with chronic inflammatory diseases but with transient inflammatory states such as sepsis. In fact, it affects the functionality of numerous organs including the pancreas, liver, adipose tissue, heart, brain and muscle tissue, leading to a systemic condition of insulin resistance [23,26]. These alterations also seem to affect the sensitivity to leptin at the level in the arcuate nucleus of the mid-basal hypothalamus with repercussions on the mechanism for regulating satiety [30]. In this context, the complex and unique endogenous mechanism underlying the resolution of the inflammation itself appears to be of great importance. Indeed, in recent years, the importance of some autacoids of lipid nature, deriving from omega-3 polyunsaturated fatty acids (PUFA) has emerged in this delicate process of limiting and resolving the inflammatory response [20,31]. In detail, these lipid mediators, called specialized pro-resolving mediators (SPMs), are the lipoxins originating from the AA, the resolvins (Rvs) D and E series, deriving from DHA and EPA, respectively, and protectins and maresine, other metabolites of DHA. They are synthesized in particular temporal windows thanks to specific heterotypic interactions of inflammatory leukocytes with the cells of the inflamed tissue, i.e., endothelial cells, epithelial cells, macrophages and platelets. Moreover, they act as regulators, both of the rhythm and of the extent of the inflammatory response [10]. In fact, SPMs act as endogenous receptor agonists at low concentrations (pM and low nM) at the level of specific trans-membrane G proteins where they determine both a downregulation of inflammatory processes and a stimulation in the resolution of exudate [32].

In the literature review by Chiang et al. [10], the numerous contributions of these autacoids in the inflammatory process are reported, including: the downregulation of cell adhesion molecules, both at the endothelial and leukocyte level, a reduction in chemotaxis and trans-endothelial migration, a lower activation of neutrophils, inhibition of both the formation and efficacy of pro-inflammatory mediators, stimulation of non-inflammatory phagocytosis of both neutrophils and apoptotic macrophages and the active release of inflammatory leukocytes. In particular, a recent review of the literature has shown that Rvs may have greater anti-inflammatory and pro-resolutive properties than the EPA and DHA precursors [10]. The delicate role of these pro-resolution molecules becomes even more important during pregnancy when persistent inflammation has a significant impact on the possible onset of maternal–fetal complications [23].

In fact, it would appear that meta-inflammation could play a pivotal role in maternal obesity and GDM, modifying the developmental programming in utero, also by influencing placental function [3].

## 3. Resolvins

### 3.1. Types and Biosynthesis

The discovery of pro-resolution lipid mediators derived from omega-3 fatty acids is relatively recent [33]. From a literature review by Bannenberg and Serhan [33], it emerged that the cyclo-oxygenase enzyme responsible for efficient oxygenation of the AA from which prostaglandin-H2 (PGH2) originates, also produces oxygenated derivatives of EPA and DHA but at a relatively low rate. Furthermore, these metabolites have shown a weak affinity for known prostaglandin receptors. However, it has been found that both cyclo-oxygenase (COX) and lipo-oxygenase (LOX) can oxygenate the omega-3s to produce intermediate oxygenated products which in turn can undergo further biotransformation. Specifically, resolvins, in humans, differ mainly according to their precursor, such as Rvs D (RvD1–6) or E series (RvE1–3) originate from DHA or EPA, respectively. For the first group there are two possible biosynthetic pathways, one is aspirin-COX-2 dependent and generates the epimeric forms activated by aspirin (AT-RvD1-AT-RvD6), while the main biosynthetic pathway is achieved through the interaction of 15-LOX and the 5-LOX. A key role is played by 17-hydroperoxy-DHA, an intermediate product of 15-LOX which acts as a limiting metabolite for the synthesis of resolvins of the D series, whose pathway marker is represented by its acid derivative, 17-hydroperoxy-DHA (17-HDHA) [34]. This mechanism seems to be established for the production of RvD1 and RvD2 and can be realized both inside mononuclear cells, such as macrophages and neutrophils, and between cells (endothelial cells–leukocytes, neutrophils–macrophages) [35,36,37].

Additionally, for resolvins deriving from EPA, or E series, two distinct biosynthetic mechanisms are possible starting from a common precursor, 18-hydroxyheicosapentaenoic acid (18-HEPE). One pathway is catalyzed by aspirin-COX-2 and 5-LOX through the interaction of endothelial cells with leukocytes while the second is aspirin-independent and begins with the guided oxygenation of EPA by cytochrome P-450 [38]. 

Both pathways lead to the formation of RvE1 and RvE2 primarily in response to increased 5-LOX concentrations during inflammation [37]. Regarding RvE3, it is mainly synthesized in eosinophils through the 12/15-LOX pathway, always starting from the common precursor 18-HEPE [39]. A simplified biosynthetic scheme of the different biosynthetic pathways is reported in Figure 1.

Valdes et al. [40] showed how these metabolic pathways generate, for the most part, unstable molecules which are therefore rapidly degraded or metabolized in vivo resulting in a short biological half-life. Moreover, the study by Mas et al. [41] found that in healthy adults the values of SPM precursors in serum and plasma were 5–10 times higher than the metabolites themselves.

### 3.2. Maternal–Fetal Concentrations

A review of the literature by Elliot et al. [10], aimed at evaluating the concentrations of SPMs in pregnant women, highlighted the basic values detected in healthy adults to be used as a comparison, despite the presence of some variables due to both the different techniques used (LC-MS/MS, liquid chromatography–tandem mass spectrometry, enzyme immune-essay), and to the potential variability of SPM values among healthy adults. Some information on the values of SPM and their precursors in pregnant women emerges from a study by Mozurkewich et al. [34] who analyzed secondary blood samples from 60 participants in the Mothers, Omega-3, and Mental Health study following dietary supplementation with EPA- and DHA-rich fish oil versus soybean oil placebo. From this analysis it emerged that, compared to placebo, supplementation with fish oil increased the precursor 17-HDHA (metabolic marker of resolvins D series), both in maternal and cord blood and that this metabolite increased significantly between enrollment and the end of gestation. It was also found that 17-HDHA values were significantly higher in umbilical cord blood than in maternal blood. With regard to placental values, Keelan et al. [42] confirm the encouraging data of murine studies [43]. In fact, this research suggests that supplementation with omega-3-rich fish oil during gestation increases the presence at the placental level of DHA (but not EPA), of the SPM precursors 17-HDHA and 18-HEPA even if the resolvins analyzed did not show a statistically significant increase. Instead, the results regarding the potential effects on gene expression do not seem to suggest any stimulatory effect on inflammation. The analysis of all these data by Elliot et al. [10] highlighted the delicate role of SPMs at the placental level, suggesting how there is a specific production and use for these mediators, although data on the very mechanism of use and receptor expression are very scarce. However, some in vitro and animal model studies performed on microglial cells [44] and cardiac fibroblast [45], respectively, have clearly demonstrated the anti-inflammatory mechanism exerted by resolvins. In detail, in these tissues, the mechanisms of resolvins in the anti-inflammation process range from the prevention of the increase in ICAM-1 and VCAM-1 protein levels, IL-6, TNF-α and IL-10 in cardiac fibroblasts to the regulation of the NFκB signaling pathway and miRNAs expression in microglia cells.

Studies on SPM concentrations in newborns are also very small and most have focused on cord blood values at the time of delivery [10]. 

Regarding the data on the SPMs’ concentration in breast milk, they derive mainly from the study by Weiss et al. [46] that analyzed the fatty acid composition of 94 human milk samples from 30 mothers in the first month of lactation using a gas chromatography–mass spectrometry (GC-MS) analysis and quantified lipid mediators through an HPLC-MS/MS (high-performance liquid chromatography–mass spectrometry). They detected the presence of RvD1 and RvE1 in breast milk, not only at higher concentrations than in the healthy volunteers, but at values known to physiologically inhibit inflammatory processes. They also found that during the first month of breastfeeding, although the SPM values remained constant, the values of alpha linolenic acid (precursor of EPA and DHA) increase, while the concentrations of DHA and 17-HDHA decrease [46]. This led the authors to hypothesize that these values respond to the high needs of DHA immediately after birth and that globally the high SPM values found could suggest an important role in the immune system, thus representing one of the possible benefits associated with breast milk. Arnardottir et al. [47] confirmed the presence of potent lipid mediators of inflammation in breast milk including D-series resolvins (RvD1–4), aspirin-activated epimeric forms (AT-RvD3) and E-series resolvins (RvE1–3). However, in the case of mastitis, the levels of these SPMs in milk were found to be lower, perhaps because of their rapid consumption due to the hyper-inflammatory state, supporting the extreme dynamism that characterizes their production.

## 4. Role in Conditions of Alteration of Maternal–Fetal Health

Although it therefore seems established that SPM levels in breast milk are dynamically regulated during inflammatory processes [10], further investigations are needed in order to assess how the alteration of SPM levels in milk correlates with inflammatory states and at-risk deliveries. Preliminary studies on the milk of women with premature birth have shown that some SPMs, including Rvd1 and Rvd2, were present at concentrations 4 times lower than the values of milk of mothers with at-term deliveries. However, the data provided by the study did not consider the possible differences due to the sampling and/or analysis methods.

From the review by Elliot et al. [10], it is evident how the production of some SPMs is regulated also according to the presence of pregnancy or perinatal pathological states. From a murine study [48], it was observed that the administration of RvE3 to pregnant mice exposed to LPS reduced the incidence of preterm labor unlike the administration of the precursor 18-HEPE which had no effect. The importance of SPMs also emerged in a study [49] which analyzed the amniotic fluid of full-term birth patients and clinical chorioamnionitis. They found that this clinical condition would seem to be characterized by a relative deficiency of SPMs. In fact, some resolvins, including RvE1 and RvD1, were not found. However, their precursors have been observed. Therefore, these results may be due both to a real deficiency of these mediators in term chorioamnionitis and caused by the labile nature of the SPMs [10,49]. Preclinical animal studies [50] highlighted the potential use of resolvins as modulators of angiogenesis and therefore a possible therapeutic use in the prevention and management of retinopathy of premature infants was hypothesized [51].

A further contribution to define more clearly the possible role of SMPs in protecting against maternal–fetal health complications comes from the study by Nordgren et al. [52]. This analysis collected data on the intake of omega-3 fatty acids from 135 mothers admitted to the delivery room with a food frequency questionnaire. The levels of RvD1 and RvD2 in both maternal plasma and cord blood were also measured and it was found that the SPM values were higher in maternal plasma than in cord blood and that the increase in these values was associated with complications. These data differ from what was observed by Mozurkewich et al. [34] regarding the values of the precursor of RvD, 17-HDHA, which were higher in maternal plasma than in cord blood. It was also found that the maternal increased intake of DHA was correlated with higher plasma levels of RvD1 and RvD2 at the time of admission to the neonatal intensive care unit. These results seem to suggest that a greater intake of omega-3 by the mother provides a favorable substrate for the formation of SPMs in the case of maternal–fetal complications and that the increase in plasma SPMs could represent a useful biomarker for these complications. Confirming the potential use of resolvins as biomarkers of pregnancy outcomes, there is the study by Aung et al. [53] which is the only metabolomics study in this specific area. It focused on the identification of associations and predictive capacity with respect to premature birth, of a large panel of metabolites (eicosanoids, immune biomarkers, oxidative stress markers and growth factors). They conducted a cross-sectional study of pregnant women in the LIFECODES birth cohort, which included 58 cases of preterm delivery, including 31 cases with spontaneous preterm delivery and/or premature rupture of membranes, 25 cases associated with placental aberrations (preeclampsia and/or intrauterine growth restriction) and 2 cases that could not be classified, compared to 115 controls. Most of the observed associations concerned spontaneous preterm birth and specifically some lipid biomarkers, including RvD1 that turned out to be among the most predictive metabolites. Furthermore, the complex statistical analysis of these data showed that lipid biomarkers proved to be the best in separating cases from controls, compared to other categories of metabolites analyzed. It also emerged that, among the different eicosanoids, those deriving from the lipoxygenase pathway showed the strongest association with preterm birth, concluding that a possible combination of lipid biomarkers may be useful to predict premature birth. In this regard, further data emerge from the very recent work by Perucci et al. [54], a longitudinal study aimed at comparing some lipid metabolites (leukotriene B4, LTB4, lipoxin A4, LXA4, RvD1) in pregnant women with risk factors for pre-eclampsia who have (*N* = 11) or have not developed (*N* = 17) this clinical condition. The analysis revealed that pregnant women with PE had both lower levels of RvD1 and an RvD1/LTB4 ratio of less than 30–34 weeks, compared to the controls. Conversely, there was an increase in RvD1 levels at 12–19 weeks in pregnant women who later developed PE. Notably, RvD1 levels were higher at 30–34 weeks than those at 20–29 weeks considering both groups of women. These data led the authors to conclude that there is a possible correlation between pregnancy outcome and gestational period in the case of RvD1. Human studies that have investigated the potential use of resolvins as clinical biomarkers in predicting pregnancy outcomes are summarized in Table 1. On the basis of the number of studies found in the literature, this potential use of resolvins is yet to be investigated. Indeed, the possible clinical relevance predicts a potential role as biomarkers of high-risk pregnancy/delivery conditions or negative neonatal outcomes, as pre-eclampsia or preterm birth, for RvD1. However, no conclusions can be drawn yet because dosages and measurement times have not been standardized according to the single pathological condition.

## 5. Role in Obesity

The contribution and therapeutic potential of resolvins in promoting an active resolution of inflammation appears to be increasingly important in pathologies characterized by high difficulty in resolving inflammatory processes and in restoring homeostasis [55,56]. This is the case of obesity, for which a growing body of scientific evidence supports the lack, at the level of immuno-metabolic tissues, of SPMs [57], although not all obese individuals have an inflammatory metabolic condition [58].

In preclinical studies on mouse models and cell cultures, the administration of exogenous resolvins has shown a promising resolving action at the level of adipose tissue and a decrease in related complications, such as non-alcoholic steatohepatitis (NASH) and insulin resistance [59,60,61,62,63].

Nevertheless, in this context, the role of resolvins still has many aspects to be defined. Originally, tissue hypoxia at the level of the adipocytes is responsible for the increase in inflammatory adipokines, which is followed by an infiltration of macrophages causing the establishment of chronic low-grade inflammation. At this level, the tissue lack of pro-resolution mediators contributes to the difficulty in resolving the inflammatory process [62]. In fact, the resolvins RvD1 and RvD2 have been shown to be responsible not only for the macrophage shift towards a pro-resolving phenotype but also for the lower migration and adhesion of monocytes to adipose tissue [62]. Their involvement in the reduction in the secretion of pro-inflammatory cytokines such as leptin, TNF-α, IL-6 and IL-1ß was also shown, together with an improvement in the expression and secretion of adiponectin [62]. Similar results emerged from the studies by Neuhofer et al. [64] who showed that obese mice showed a rapid reduction, compared to the controls, of both SPMs and precursors deriving from DHA in white adipose tissue within four days of consuming a high-fat diet. Furthermore, treatment with 17-HDHA decreased the expression of inflammatory cytokines, increased the expression of adiponectin and improved glucose tolerance in parallel with insulin sensitivity in obese mice. These data underline the delicate role of resolvins in the inflammation of the adipose tissue in obesity.

In this complex condition of altered production of obesity-related SPMs in various immuno-metabolic tissues, the consequences at the level of the central nervous system must be added, with repercussions in terms of energy storage. In this regard, diet-induced hypothalamic inflammation is an important mechanism that leads to dysfunction of the neurons involved in the control of body mass. Specifically, the study by Pascoal et al. [65] showed a reduction in RvD2 at the hypothalamic level in obese mice.

Since pregnancy is already characterized by mild but significant inflammatory activity in physiological conditions [66,67], when complicated by obesity the probability of a persistent inflammatory state increases, with consequent multiple repercussions that add up to the complications associated with acute inflammation [67,68], as show in Figure 2. It is in fact known that elevated levels of C-reactive protein, a systemic marker of inflammation, are related to preeclampsia, preterm delivery, pregnancy loss and fetal growth restriction [69]. Other systemic indicators of inflammation, including IL-1ß, have recently been associated with impaired fetal growth and adverse birth outcomes [70]. Furthermore, a recent study conducted by Han et al. [71] showed that a maternal inflammatory state, both chronic and acute, is related to alterations in neurodevelopment that can lead to autistic spectrum disorder (ASD) and the attention deficit hyperactivity disorder syndrome (ADHD).

In addition, in the case of obesity, the imbalance of inflammatory cytokines and the consequent high levels of circulating leptin together with a condition of leptin resistance [72], can also affect the birth itself [68]. In fact, the physiological inhibitory effect of leptin on uterine contractility can contribute to the high frequency of dysfunctional labor associated with maternal obesity and the consequent high rates of caesarean section [73]. Furthermore, the consequences of this altered inflammatory state can also affect the delicate glycemic control during pregnancy, in fact Resolvin D1 is involved in insulin sensitivity due to its influence on insulin cell signaling pathways and related inflammatory pathways [74], a condition that could support the high incidence rate of gestational diabetes in pregnancies characterized by maternal obesity. 

As for the underlying causes of the SPM deficiency in the adipose tissue, they may be different and have not yet been fully clarified. Previous studies have shown an upregulation of some key enzymes of the metabolic pathways that leads to the inactivation of SPMs, such as that of PG-dehydrogenase/eicosanoid oxidoreductase [62] and inefficient biosynthesis [75]. It should also be noted that this deficiency in obesity seems to affect other immune-metabolic tissues in addition to the adipose tissue, including muscle tissue [75]. By contrast, the study by Echeverría et al. [76] revealed unusual results, showing that although hepatic EPA and DHA content was low in obese mice, compared to the controls, the level of some resolvins, including RvE1, RvE2, RvD1 and RvD2, was likely increased as a reaction to pre-existing chronic low-grade inflammation. More recent studies have instead confirmed the altered balance between resolvins of the D series and pro-inflammatory mediators in favor of the latter also in the peripheral blood leukocytes of obese subjects, supporting a multi-organ inflammatory condition characteristic of obesity [77]. The Bashir et al. study [78] confirmed an overall decrease in resolvins in the adipose tissue of the mouse model (RvE1, RvE2, RvD2, RvD3, RvD5), with the exception of RvD6. In addition, the in vitro analysis performed by López-Vicario C et al. [77] seems to confirm that the failure to resolve the inflammation detected in the peripheral blood leukocytes of obese subjects should be attributed precisely to an ineffective biosynthesis rather than to an impaired degradative metabolism [62]. In fact, a decreased expression of the 15-LOX enzyme has emerged and supplementation with 17-HDHA, a by-product of this enzyme, has been shown to be effective. These results suggest an alternative use of omega-3s as an anti-inflammatory nutritional therapy in the future [77] and confirm what has been demonstrated by the preclinical studies by Neuhofer et al. [64].

## 6. Correlation with SARS-CoV-2 Infection

Obesity seems to be associated with a greater susceptibility to infections [79,80,81]. In detail, the scientific literature mainly reports an increased production of inflammatory cytokines and chemokines which seems to be among the causes of a reduced function of the cells of the immune system that affects the innate and acquired response to both bacterial and viral infections [81]. The microbiome/virome imbalance in the intestine also has repercussions on the alteration of the efficiency of the immune system [81]. In this context, the state of fatty acids has already proved to be decisive as it has been found to be a factor influencing humoral immunity, potentially through a mechanism mediated by SPM [79].

Although obesity was not initially mentioned frequently among the clinical risk factors for SARS-CoV-2 infection, to date, scientific research has produced several new data on this matter [81]. The study by Simonnet et al. [82] showed a high frequency of obesity among patients admitted to intensive care for SARS-CoV-2, also revealing an increase in the severity of the disease with the increase in BMI. Liu et al. [83] highlighted the association between high BMI and severity of the infection in COVID-19-positive patients. The correlation between high BMI and lower survival rate emerged from the analysis by Peng et al. [84]. In addition, Lighter et al. [85] found, even in younger subjects, a correlation between obesity and a negative COVID-19 prognosis, highlighting how patients under the age of 60 with a BMI between 30 and 34 were more likely to be admitted to the ICU than individuals with a BMI < 30.

The obesity-related lack of SPMs seems to be decisive in this scenario as well, as it appears to be one of the causes associated with the higher incidence of complications and negative outcomes of SARS-CoV-2 infection [52,81]. This means SPMs are involved in key mechanisms through which infection leads to the cytokine storm and the altered and dysregulated lung response [57]. Indeed, Costela-Ruiz et al. [86] mainly showed a correlation between the overproduction of pro-inflammatory cytokines, such as IL-1, IL-6, interleukin-12 (IL-12), interferon gamma (IFN-γ) and TNF-α, especially in the lung tissue, and worse prognosis in severe cases of SARS-CoV-2 infection. In this regard, the role of RvD1 and RvD2 in reducing the secretion of pro-inflammatory cytokines including IL-6 and TNf-α [62] has already been discussed. The overproduction of IL-6 and TNF-α has also been associated with a dysregulation of certain fat-resident regulatory T cells (Treg), and in particular with the promotion of Th-17 immunity (T cell sub lineage), in turn correlated with the mechanisms involved in the higher incidence and mortality from COVID-19 in obese patients [81].

In pregnancy, this situation is further complicated, probably due to several factors. Among these, the particular cardiopulmonary conditions (acute cardiopulmonary conditions) present in the second and third trimester of pregnancy [87] and the specific immune phenotype of the pregnant woman [88,89]. The latter, characterized by both an altered Th1/Th2 ratio [89] and a variation of immune priorities, with a strengthening of innate immune barriers and a concomitant reduction in adaptive/inflammatory immunity in the later stages of pregnancy [88]. The greater morbidity and mortality in pregnancy from various infections is known in the literature [81], including all the influenza pandemics documented to date [90,91,92] and SARS [93], in the case of overweight, pre-pregnancy obesity and obesity. The data available to date on the SARS-CoV-2 infection in pregnancy seem to confirm this trend. Indeed, from the meta-analysis by Khalil et al. [94], 38.2% of 2567 pregnant women with SARS-CoV-2 symptoms were obese. Furthermore, there are the data collected by eight United States Care Centers which showed that pre-pregnancy obesity and gestational diabetes were prevalent in pregnant women hospitalized for COVID-19 compared to all the pregnant women who tested positive at the time of delivery [95]. Moreover, in the analysis of the literature by Lapolla et al. [96], the data reported by the Vaccine Safety Datalink relating to the surveillance of COVID-19 hospitals from 1 March to 30 May 2020 highlighted that conditions such as pre-pregnancy obesity and gestational diabetes were more common among pregnant women hospitalized for COVID-19 than among pregnant women hospitalized for obstetric reasons.

## 7. Utility of Dietary/Supplement Modulation of Resolvin Concentrations?

Surely interventions on eating habits can represent the first step to ensure an adequate concentration of resolvins, although this approach could reveal some critical issues. First of all, it provides the vehicle of precursors (EPA and DHA) less active than resolvins [22]. Secondly, there is no possibility of compensating for any enzymatic alterations of the biosynthetic type (15-LOX) [75,77] and/or degradative type (PG-dehydrogenase/eicosanoid oxidoreductase) [62] responsible for resolvin deficiency in some particular conditions such as obesity. In fact, although the data available to date suggest that placental and blood levels of SPMs can be modulated during pregnancy through the intake of omega-3 fatty acids and that SPMs play a key role in the benefits associated with the intake of LC- PUFA [10,34], in the case of maternal obesity, the production of resolvins could be altered. Despite this, the contribution of a correct and well-balanced diet appears essential for several reasons. Primarily the negative effects of a high-fat diet on the production of SPMs and its precursors have already been highlighted in the animal model [64]. Moreover, the data available on the Western diet suggest that its content in omega-3 fatty acids is low [97] while the intake of omega-6 is excessively high, with consequent possible enzymatic competition [98,99]. Finally, these eating habits have been included among the environmental factors that contribute to the high morbidity associated with lung diseases in the context of obesity [100].

There is also the possibility of contributing to the achievement of an adequate production of resolvins through food supplementation, which have proven their effectiveness under certain conditions. In fact, the usefulness of this practice emerged from the previously discussed study by Keelan et al. [42], which highlighted the effectiveness of supplementation with omega-3-rich fish oil during gestation in relation to the increase in DHA, 17-HDHA and 18-HEPA at the placental level. In addition, there is the recent contribution by Souza et al. [101], who analyzed the relationship between omega-3 supplementation and peripheral SPMs concentrations, demonstrating how supplementation with an enriched marine oil leads to an increase in concentrations of SPMs in peripheral blood in a time-dose dependent manner, with important repercussions at the level of the immune system. In pregnant women, reaching the average daily intakes for DHA of 300 mg/day (triple quantity compared to healthy adults), a preventive value recently proposed by Koletzko et al. [17]. These are the appropriate rates according to the guidelines of the European Food Safety Authority (EFSA) that has established an adequate daily intake during pregnancy of 250 mg of EPA plus DHA (adult requirement based on cardiovascular considerations) plus an additional 100–200 mg of preformed DHA to compensate for oxidative losses of maternal dietary DHA and accumulation of DHA in fetal/newborn body fat [17]. This is the minimum quantity for preventive purposes. Indeed, higher intakes (600–800 mg DHA/day) may guarantee greater protection than lower doses, but direct comparative dose–effect evaluations are not available [17]. However, the correlation with correct dietary habits is essential due to possible enzymatic competition that could affect the effectiveness of the supplement itself in the presence of highly unbalanced dietary intake [97,98,99].

Regarding a targeted and personalized dietary supplementation through the direct administration of resolvins, it has been found to be effective in reducing obesity-related complications in several preclinical studies on animal models, as previously discussed [59,60,61,62,63]. The efficacy of RvE3 was also tested in pregnant mice exposed to LPS against the prevention of preterm birth in the study by Yamashita et al. [43]. A further contribution comes from López-Vicario C et al. [77], who highlighted the effectiveness of supplementation with 17-HDHA following the detection of an altered biosynthesis of resolvins in leukocytes of obese subjects. Then, there is the study by Quiros et al. [102], which by conveying RvE1 in synthetic polymer nanoparticles, demonstrated its effectiveness in healing wounds in a model of the human intestine. Further data on the need for innovative and increasingly personalized nutritional therapy are provided by Pal et al. [103], who highlighted the need for a personalized administration of RvE1 according to the specific metabolic profile. In fact, they observed that the beneficial effects of EPA on hyperinsulinemia and hyperglycemia are due in part to the activation, dependent on the host genetic manner of the receptor coupled to the G protein ERV1/ChemR23, by RvE1.

Nevertheless, for this possible therapeutic approach, the methods of administration still need to be carefully evaluated [57], also due to the short biological half-life of some compounds [40].

## 8. Conclusions

The scientific literature available to date seems to suggest that maternal omega-3 intake may influence the presence of resolvins, both in maternal blood and at the placental level. There is also a modulation in the production of resolvins during the course of gestation and a correlation between maternal–fetal complications and the amounts of these molecules. In this context, obesity, a resolvin deficiency, during pregnancy increases the probability of a persistent inflammatory state, the repercussions of which are manifold and add up to the complications associated with acute inflammation [65].

Based on these premises, the usefulness of metabolomics in this field appears clear, a technology capable of providing a unique phenotypic portrait of an organism by virtue of its ability to directly measure multiple metabolites within complex biological systems. Indeed, through the metabolome it is possible to observe the numerous and complex interactions between the mother, the placenta and the fetus in order to identify specific biomarkers useful in the prediction, diagnosis and monitoring of the various obstetric conditions [104]. Moreover, metabolomics measure global sets of low molecular weight metabolites thus providing a “snapshot” of the metabolic status of an organism in relation to genetic variation and external stimuli [105]. This feature could allow us to overcome the short half-life of resolvin-related products, which to date seems to be the main problem in clinical practice.

Further investigations are therefore needed, with a view on preventive medicine, in order to evaluate not only the possible use of some resolvins as biomarkers of maternal–fetal outcomes but also to establish adequate integration values in pregnant women with omega-3 fatty acids or with more active derivatives that guarantee optimal SPM production under risky conditions.

## Figures and Tables

**Figure 1 nutrients-14-01662-f001:**
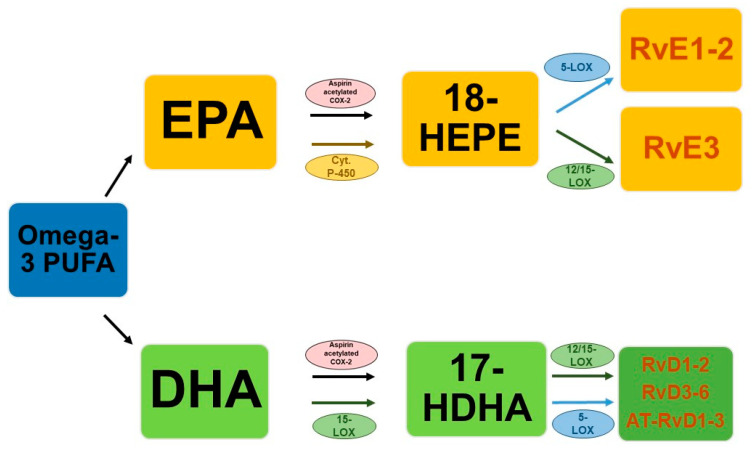
Simplified scheme of the biosynthesis of resolvins.

**Figure 2 nutrients-14-01662-f002:**
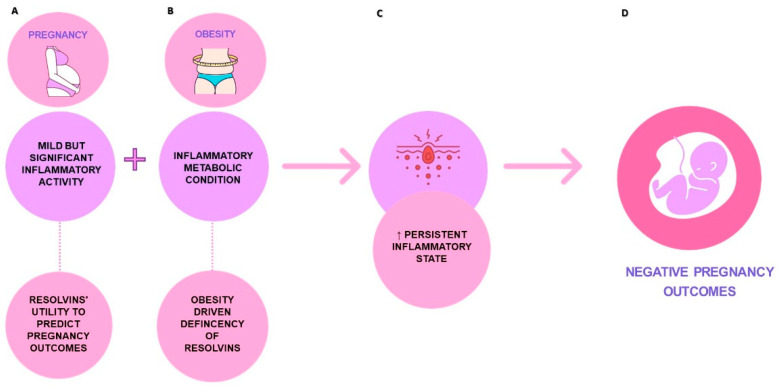
Pregnancy and obesity involvement in inflammatory state and resolvins’ correlation: (**A**) mild but significant inflammatory activity in physiological pregnancy where resolvins could be useful biomarkers for pregnancy and neonatal outcomes (**B**) in pregnancies complicated by obesity the probability of a persistent inflammatory state increases also due to the resolvins’ deficiency, (**C**) consequences are a persistent inflammatory state and multiple repercussions that add up to the complications associated with acute inflammation and (**D**) subsequent negative pregnancy outcomes.

**Table 1 nutrients-14-01662-t001:** Studies analyzing resolvins’ utility in clinical settings to predict pregnancy outcomes.

Authors, Year	Patients	Sample	Technique	Main Results	Clinical Significance
Nordgren et al. [52], 2019	136 mothers and 138 infants at the time of delivery.	Maternal and cord plasma.	Commercially available enzyme immunoassays	↑ SPM in maternal versus cord plasma ↑ SPM levels associated with at-risk outcomes ↑ maternal DHA intake associated with ↑ maternal plasma RvD1 e RvD2	Increased n-3 fatty acid intake may provide increased substrate for the production of SPM during high-risk pregnancy/delivery conditions Increased maternal plasma SPM could serve as a biomarker for negative neonatal outcomes
Aung et al. [53], 2019	58 cases of preterm birth and 115 controls	Blood samples	6490 Triple Quadrupole mass spectrometer	RvD1 were among the most predictive markers Lipid biomarkers were the best in separating cases from controls Eicosanoids resulting from the lipoxygenase pathway showed the strongest association with preterm childbirth	The combination of lipid biomarkers may have good utility in clinical settings to predict preterm birth Most of the observed associations relate to spontaneous preterm birth
Perucci et al. [54], 2021	28 pregnant Brazilian women, 11 women developed PE and 17 remained normotensive	Blood samples	Competitive enzyme-linked immunosorbent assays (ELISA)	↓ RvD1 levels and ↓ RvD1/LTB4 ratio at 30–34 weeks in pregnant women with PE In pregnant women who developed PE ↑RvD1 levels at weeks 12–19 and ↑ LXA4 and RvD1 levels at 30–34 weeks than those at 20–29 weeks	Potential use of lipid mediators (RvD1) as clinical markers for PE development

Abbreviations: ↑ increase, ↓ decrease, PE, pre-eclampsia.

## Data Availability

Not applicable.

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
