# Peer review of "Resolvins’ Obesity-Driven Deficiency: The Implications for Maternal–Fetal Health"

_nutrients, 2022, doi:10.3390/nu14081662_

Round 1
Reviewer 1 Report
Current review article described the potential role of Resolvins, one of the specialized pro-resolving mediators (SPMs) deriving from omega-3 essential fatty acids, as the biomarker in maternal-fetal outcomes. Please conduct the concerns below.
- Obesity in pregnancy and gestational diabetes are frequently associated with a chronic low-grade inflammation, defined as meta-inflammation. Effects of COVID-19 pandemic or omega-3 fatty acids on meta-inflammation were not conducted.
- The persistent inflammation during pregnancy has a significant impact on the onset of maternal-fetal complications. Associations with meta-inflammation were not introduced.
- SPMs in milk were found to be lower than breast milk that needs a possible reason even speculation in line 231.
- Development of resolvins as clinical biomarkers in predicting pregnancy outcomes must describe in detail. Which one is more reliable?
- In mice, administration of exogenous resolvins ameliorated non-alcoholic steatohepatitis (NASH) and insulin resistance (IR). Is it also observed in humans?
- Figure 2 needs the legends to explain in detail.
- In line 400, obesity-related lack of SPMs that needs reference(s).
- In line 465, proposed by Koletzko et al. that needs reference(s) too.
- In pregnant women, the daily intakes of DHA at 300 mg were suggested. Please discuss the backgrounds for this triple quantity compared to healthy adults.
- The short half-life of resolvin-related product seems the main problem in clinical practice. How to solve this problem in advance?
- Conclusions seem too complicated and hard to know the key points of current report. Please improve it to be brief with novelty.
- Effective dose of the most reliable resolvin-related product is most helpful to readers.
Author Response
Rev. 1
Current review article described the potential role of Resolvins, one of the specialized pro-resolving mediators (SPMs) deriving from omega-3 essential fatty acids, as the biomarker in maternal-fetal outcomes. Please conduct the concerns below.
- Obesity in pregnancy and gestational diabetes are frequently associated with a chronic low-grade inflammation, defined as meta-inflammation. Effects of COVID-19 pandemic or omega-3 fatty acids on meta-inflammation were not conducted.
Thank you for this suggestion, we have modified the text accordingly.
- The persistent inflammation during pregnancy has a significant impact on the onset of maternal-fetal complications. Associations with meta-inflammation were not introduced.
Thanks for this comment, we changed the text taking it into account.
- SPMs in milk were found to be lower than breast milk that needs a possible reason even speculation in line 231.
Thank you for this suggestion we provided a speculation in the text: SPMs in milk were found to be lower than breast milkmaybe because of to their rapid consumption due to the hyper-inflammatory state caused by mastitis.
- Development of resolvins as clinical biomarkers in predicting pregnancy outcomes must describe in detail. Which one is more reliable?
Thanks for this comment. We provided further details according to your suggestion in the text: “As emerge from the number of studies this potential use of resolvins has yet to be investigated. The possible clinical relevance that emerges from all these studies predicts a potential role as biomarkers of high-risk pregnancy/delivery conditions or negative neonatal outcomes, as pre-eclampsia or pre-term birth, for RvD1. However, no conclusions can be drawn yet, also because dosages and measurement times have not been standardized according to the single pathological condition.”
- In mice, administration of exogenous resolvins ameliorated non-alcoholic steatohepatitis (NASH) and insulin resistance (IR). Is it also observed in humans?
Thanks for this clarification, according to our revision of the literature there is no human studies on this topic.
- Figure 2 needs the legends to explain in detail.
We have provided the details in the text
- In line 400, obesity-related lack of SPMs that needs reference(s).
Thank you for this comment and we are sorry for not being clear enough, however the reference regarding the association between covid-19 and obesity related resolvins’ deficiency is the same as stated at the end of the sentence.
- In line 465, proposed by Koletzko et al. that needs reference(s) too.
Thanks for this comment, we apologize for this inaccuracy. We have added it.
- In pregnant women, the daily intakes of DHA at 300 mg were suggested. Please discuss the backgrounds for this triple quantity compared to healthy adults.
Thanks for this comment, we have discussed the backgrounds for this triple quantity compared to healthy adults in the text accordingly.
- The short half-life of resolvin-related product seems the main problem in clinical practice. How to solve this problem in advance?
Thank you for this suggestion, we have speculated in the conclusions the possibility of solving this problem in clinical practice. “Moreover, metabolomics measure global sets of low molecular weight metabolites thus providing a "snapshot" of the metabolic status of an organism in relation to genetic variation and external stimuli and this feature could allow to overcome short half-life of resolvin-related product seems that to date, it seems the main problem in clinical practice.”
- Conclusions seem too complicated and hard to know the key points of current report. Please improve it to be brief with novelty.
Thanks for this comment, we have tried to summarize the conclusions by highlighting the main points and innovations.
- Effective dose of the most reliable resolvin-related product is most helpful to readers.
Thank you for this suggestion, however we did not discuss the specific dosages since the analysis of the correlation between maternal fetal health and resolvins is a very new topic: there are only three studies in literature, according to our knowledge, one of which is non-quantitative metabolomics. Further studies are needed to define this dose.
Reviewer 2 Report
The review discusses extensively on clinical observations of SPM and obesity, and maternal-fetal health. This is a novel aspect which are not found in other reviews of SPMs.
Comments
- The review article discusses very nicely the observed clinical effects of dietary supplements such as omega-3 oils on increasing SPMs and their precursors. In section 3.2, line 207 stated that after supplementation, there was no statistical increase in resolvins. This statement brings up questions on the mechanisms of resolvins in the anti-inflammation process, which can be further supported by other in vitro works such as those below. Hope the authors can consider incorporating a little more info on the biological mechanisms of the resolving anti-inflammation process.
Rey, Charlotte, et al. "Resolvin D1 and E1 promote resolution of inflammation in microglial cells in vitro." Brain, behavior, and immunity 55 (2016): 249-259.—” BV2 cells were pre-incubated with RvD1 or RvE1 before lipopolysaccharide (LPS) treatment. RvD1 and RvE1 both decreased LPS-induced proinflammatory cytokines (TNF-α, IL-6 and IL-1β) gene expression, suggesting their proresolutive activity in microglia. However, the mechanisms involved are distinct as RvE1 regulates NFκB signaling pathway and RvD1 regulates miRNAs expression.”
Salas-Hernández, Aimeé, et al. "Resolvin D1 and E1 promote resolution of inflammation in rat cardiac fibroblast in vitro." Molecular Biology Reports 48.1 (2021): 57-66.—" RvD1, but not RvE1, prevents the LPS-induced increase of IL-6, MCP-1, TNF-α, and IL-10.”
- It is stated in Section 4 (lines 248-249) that “In fact, some resolvins including RvE1 and RvD1 were not found. However, their precursors have been observed.”. Are there any explanations why were the precursors not converted to the SPMs since the precursors were found? Was there a defect in the relevant genes/enzyme that led to the precursors not being converted to the active SPMs? Was a knockout/knockdown experiment performed to show that this could actually lead to the same end point of missing resolvins and increased incidence of chorioamnionitis or related inflammation? Such would be interesting works/future works to suggest in this review, in case there is a common genetic defect which predisposes individuals to lack of SPM.
- It was highlighted in the abstract that metabolomics is useful in the field, but there were only a few cases of metabolomics cited in the article. Perhaps authors can elaborate more on how metabolomics can be used to evaluate the role of SPMs in the diseases discussed by citing more examples if available. This will also be useful in encouraging researchers to try out metabolomics for SPM studies.
Minor comments
Line 157: Typo: “Furthermore, these 1metabolites have shown…”—extra “1” before metabolites
Line 195: “PMS” should be SPM?
Author Response
Rev. 2
Comments and Suggestions for Authors
The review discusses extensively on clinical observations of SPM and obesity, and maternal-fetal health. This is a novel aspect which are not found in other reviews of SPMs.
Comments
- The review article discusses very nicely the observed clinical effects of dietary supplements such as omega-3 oils on increasing SPMs and their precursors. In section 3.2, line 207 stated that after supplementation, there was no statistical increase in resolvins. This statement brings up questions on the mechanisms of resolvins in the anti-inflammation process, which can be further supported by other in vitro works such as those below. Hope the authors can consider incorporating a little more info on the biological mechanisms of the resolving anti-inflammation process.
Thanks for this suggestion, we have added more information concerning the biological mechanisms of the resolving anti-inflammation process using the references provided.
Rey, Charlotte, et al. "Resolvin D1 and E1 promote resolution of inflammation in microglial cells in vitro." Brain, behavior, and immunity 55 (2016): 249-259.—” BV2 cells were pre-incubated with RvD1 or RvE1 before lipopolysaccharide (LPS) treatment. RvD1 and RvE1 both decreased LPS-induced proinflammatory cytokines (TNF-α, IL-6 and IL-1β) gene expression, suggesting their proresolutive activity in microglia. However, the mechanisms involved are distinct as RvE1 regulates NFκB signaling pathway and RvD1 regulates miRNAs expression.”
Salas-Hernández, Aimeé, et al. "Resolvin D1 and E1 promote resolution of inflammation in rat cardiac fibroblast in vitro." Molecular Biology Reports 48.1 (2021): 57-66.—" RvD1, but not RvE1, prevents the LPS-induced increase of IL-6, MCP-1, TNF-α, and IL-10.”
- It is stated in Section 4 (lines 248-249) that “In fact, some resolvins including RvE1 and RvD1 were not found. However, their precursors have been observed.”. Are there any explanations why were the precursors not converted to the SPMs since the precursors were found? Was there a defect in the relevant genes/enzyme that led to the precursors not being converted to the active SPMs? Was a knockout/knockdown experiment performed to show that this could actually lead to the same end point of missing resolvins and increased incidence of chorioamnionitis or related inflammation? Such would be interesting works/future works to suggest in this review, in case there is a common genetic defect which predisposes individuals to lack of SPM.
Thank you for this comment and we apologize that we were not sufficiently clear. Nevertheless, one possible explanation for the presence of some resolvins precursors and not pro-resolution derivatives is provided in the following sentence: “Therefore, these results may be due both to a real deficiency of these mediators in term chorioamnionitis and caused by the labile nature of the SPMs” This explanation derives both from the authors’ discussion of the work and from subsequent literature reviews.
- It was highlighted in the abstract that metabolomics is useful in the field, but there were only a few cases of metabolomics cited in the article. Perhaps authors can elaborate more on how metabolomics can be used to evaluate the role of SPMs in the diseases discussed by citing more examples if available. This will also be useful in encouraging researchers to try out metabolomics for SPM studies.
Thank you for this suggestion, we have added further information regarding the potential application of metabolomics in the conclusions, however, this topic is so novel that we found no other studies on the potential clinical use of resolvins for maternal fetal health.
Minor comments
Line 157: Typo: “Furthermore, these 1metabolites have shown…”—extra “1” before metabolites
Thank you for this suggestion, we have corrected it.
Line 195: “PMS” should be SPM?
Thank you for this suggestion, we have corrected it.